# DNA sequencing of whole human cytomegalovirus genomes from formalin-fixed, paraffin-embedded tissues from congenital cytomegalovirus disease cases

Kathy K. Li[ID][¤a], Nicolás M. Suárez[¤b], Salvatore Camiolo[¤c], Andrew J. Davison[ID], Richard J. Orton*

MRC-University of Glasgow Centre for Virus Research, Sir Michael Stoker Building, Garscube Campus, Glasgow, United Kingdom

¤a Present addresses: Regional Virus Laboratory, Belfast Health and Social Care Trust, Belfast, United Kingdom.
¤b Present addresses: Departamento de Bioquímica, Biología Molecular, Fisiología, Genética e Inmunología, Universidad de Las Palmas de Gran Canaria, Las Palmas de Gran Canaria, Spain.
¤c Present addresses: BioClavis Ltd, 201 Dumbarton Road, Glasgow , United Kingdom.
* richard.orton@glasgow.ac.uk

## Abstract

### Background

Congenital cytomegalovirus disease (cCMV) is uncommon but can be severe. Investigations of the role of genome sequence variation in the causative virus (human cytomegalovirus, HCMV) in clinical outcome have to date depended on small sample numbers derived from fresh tissues. Extensive formalin-fixed, paraffin-embedded (FFPE) cCMV biorepositories established worldwide potentially provide much larger sample numbers for future investigations. However, there are no published reports of sequencing whole HCMV genomes from such material.

### Objective

To sequence whole HCMV genomes from cCMV FFPE material

### Study design

Sixteen FFPE samples of foetal kidney or placental tissue were processed from ten cCMV cases in foetuses or neonates. Two commercial kits for extracting DNA from FFPE material were evaluated, HCMV DNA was enriched in the extracts, and the samples were sequenced on the Illumina platform. The sequence read datasets were analysed by genotyping, genome assembly and variant calling using a published software pipeline.

**Data availability statement:** Read datasets and HCMV genome sequences are available from NCBI BioProject PRJNA1181764, NCBI Sequence Read Archive (SRA) and NCBI GenBank, respectively, under the accessions listed in Table 2 of the manuscript. URLs included in submission.

**Funding:** K.K.L. received funding award from Medical Research Council, grant number MC_ST_00034. A.J.D. received funding from Medical Research Council, grant numbers: MC_UU_12014/3 and MC_UU_12014/12 and from Wellcome, grant number 204870/Z/16/Z. Funder websites are as follows: https://www.ukri.org/councils/mrc/ and https://wellcome.org/. Neither of the funding bodies had any role in the study design, data collection and analysis, decision to publish, nor preparation of the manuscript.

**Competing interests:** The authors have declared that no competing interests exist.

## Results

Whole HCMV genomes were sequenced for five cases using either DNA extraction kit.

## Conclusions

Sequencing whole HCMV genomes from cCMV FFPE material is feasible. This potentially facilitates future studies of the effects of HCMV variation on the clinical outcome of cCMV.

## Introduction

Congenital cytomegalovirus disease (cCMV) is the most common non-genetic cause of sensorineural hearing loss and neurodevelopmental delay [1]. The role of variation in the causative virus (human cytomegalovirus, HCMV) in clinical outcome has been investigated in several studies [2]. These studies focused on hypervariable HCMV genes in order to determine whether particular genotypes are associated with virulence in single-strain infections, and whether multiple-strain infections are more virulent than single-strain ones. However, as cCMV affects only 1 in 100–150 live births [3], access to clinical samples is limited. Biorepositories of formalin-fixed, paraffin-embedded (FFPE) tissues commonly collected in pathology departments thus offer a resource for future studies.

Archived placental FFPE samples have proved useful as an adjunct in diagnosing infants asymptomatic of cCMV at birth, and some studies have used such samples to detect HCMV by immunohistochemistry or PCR amplification of short genomic fragments [4,5]. However, to our knowledge, no published work has involved sequencing whole HCMV genomes from FFPE material. This is due largely to the difficulty of recovering DNA of sufficient quality [6], as formalin adversely affects nucleic acid integrity.

### Objective

To assess the feasibility of sequencing whole HCMV genomes from archived FFPE material.

### Materials and methods

Sixteen FFPE samples of placental or foetal kidney tissue from ten cCMV cases (2008–2018) were retrieved from the pathology archive at Birmingham Women's Hospital, UK. The associated pseudonymised data were collected by a member of the primary care team on 18 September 2018. These samples, labelled with delinked reference numbers, were sent with the pseudonymised data to the MRC-University of Glasgow Centre for Virus Research for sequencing. Ethical approval was granted by the Health Research Authority Research Ethics Committee (HRA REC reference 18/LO/1441; R&D number 18/BW/NNU/NO17; 31 August 2018), and consent for future research on excess samples was obtained at the time of sampling by the primary

care team for tissues retained in the Birmingham biorepository. The authors had no access to patient-identifiable data during or after the study. The cases included five from intra-uterine death, two from termination of pregnancy, one from miscarriage, and two from neonatal death (Table 1).

Two kits for extracting DNA from FFPE material via different methodologies were assessed: one using a paramagnetic bead-based approach (FormaPure DNA extraction and purification kit, Beckman Coulter) and the other using spin-column technology (GeneRead DNA FFPE kit, QIAGEN). DNA load in the extracted samples was determined using a Qubit fluorometer (ThermoFisher Scientific), and HCMV and human DNA loads were determined by qPCR targeting the HCMV UL97 [7] and human *FOXP2* genes [8], respectively (S1 Table). Only samples with an HCMV load >100 IU/μL were processed for sequencing. The extracts were enriched for HCMV DNA by hybridisation-based capture [9] and sequenced on the Illumina platform. GRACy, a software pipeline for determining HCMV genome sequences from Illumina data [10], was used to analyse each sequence read dataset by read filtering, genotyping, genome assembly and variant (single nucleotide polymorphism; SNP) calling.

The read filtering module removed human reads, trimmed adapters and low-quality nucleotides, and removed duplicate reads.

The genotyping module enumerated sequence motifs in the filtered datasets that were specific to the genotypes of 13 hypervariable HCMV genes, thus allowing the number of HCMV strains in a sample to be estimated without requiring genome assembly. For each dataset, a more stringent threshold than that used for fresh clinical samples, akin to that used in human genetics for FFPE samples, was applied to assign genotypes to each gene: > 100 reads representing >5% of reads detected for all genotypes of that gene [11,12,13,14]. The number of strains was then registered as being the greatest number of genotypes detected for at least two genes, with a requirement for consistent assignment of genotypes across datasets from the same case. In addition, this module determined whether the combination of 13 genotypes for each dataset was represented among a large collection of published HCMV genome sequences.

The genome assembly module produced a draft HCMV sequence from each dataset. The original datasets for each case were then combined, processed using Trim Galore v.0.4.0 (https://www.bioinformatics.babraham.ac.uk/projects/trim_galore/), and aligned to the best draft assembly for that case using Bowtie 2 v2.4.2 [15] with the --local parameter. The read alignment was visualised using Tablet v1.21.02.08 [16], and improvements were implemented manually to yield the final sequence. Read coverage was determined by aligning each dataset to the final sequence. The variant calling module applied a threshold similar to that used commonly in human somatic allelic calling: a frequency of 5% [11,14] and a coverage of 50 reads/nt.

## Results

DNA extracts of sufficient quality for sequencing were obtained from all cases but case 660 (S1 Table). These included 11 extracts from nine cases using the FormaPure kit and eight extracts from six cases using the GeneRead kit. Extracts prepared using the GeneRead kit contained more DNA but had higher A260/280 ratios (indicative of residual RNA) than those prepared using the Formapure kit (S1 Fig.). However, there was no significant difference between the two kits in the quality of the HCMV sequence data generated, as assessed from the average coverage depth of a reference HCMV genome (S1 Fig.).

Genotyping was carried out for 19 datasets from 12 FFPE samples from nine cCMV cases (Fig 1). Analysis of three datasets (124R_fp, 35R_gr and 70P_fp) did not meet threshold requirements probably because of a combination of low DNA load and low proportion of HCMV DNA (S1 Table). Analysis of the remaining 16 datasets indicated that eight cases involved a single HCMV strain and one (case 70) may have involved one or more additional minor strains. None of the combinations of 13 genotypes for each dataset was represented among published HCMV genome sequences. This is consistent with prior evidence that, due to intrastrain recombination during HCMV evolution, vast numbers of genotype combinations exist among natural strains [12,17,18].

**Table 1. Pseudonymised metadata from cCMV cases used in this study.**

| Case no. | Sample age[a] | Tissue[b] | Source[c] | Individual age[d] | Post-mortem findings | Placental histopathology | Maternal infection | Antenatal findings |
|---|---|---|---|---|---|---|---|---|
| 184 | 1 | P, R | IUD | 20 w | Intra-uterine growth retardation, liver fibrosis, encephalitis/necrosis, inclusions in lung, liver, kidney, testis, thyroid and brain | Large, mild chronic villitis, abundant inclusions | Unknown | None |
| 70 | 2 | P, R | TOP | 38 w | Cerebral necrosis/meningoencephalitis, inclusions in lungs, pancreas, kidneys and brain | Normal size, mild plasmacytic villitis, plasma cells, occasional inclusions | Primary | Ventriculomegaly |
| 150 | 2 | P | IUD | 20 w | Not available | Necrotising chronic villitis, plasma cells, inclusions | Unknown | Small for gestational age, echogenic bowel |
| 413 | 2 | P, R | TOP | 21 w | Cerebral necrosis/meningoencephalitis, polymicrogyria, vermis and corpus callosum present, splenomegaly, inclusions in lung, liver, kidney, pancreas, thyroid, adrenals | Small, severe plasmacytic villitis, occasional inclusions | Unknown | Echogenic bowel; there was also thought to be vermian agenesis, an indistinct cavum septum pellucidum, raising the possibility of the absence of the corpus callosum and dilated cerebral ventricles |
| 35 | 5 | P, R | IUD | 22 w | Hydrops, large liver, dilated heart, pulmonary hypoplasia, scanty inclusions, normal brain | Large, hydropic, diffuse plasmacytic villitis, inclusions | HCMV IgG-positive | Dilated heart, intra-uterine growth retardation |
| 239 | 5 | P, R | IUD | 34 w | Microcephaly, hypoplastic corpus callosum and vermis, abnormal gyration, cholestasis, large spleen, inclusions in lung, liver, kidney, pancreas, brain | Small, plasmacytic villitis, numerous inclusions | Unknown | Growth restriction, borderline ventriculomegaly, posterior callosal deficiency, delayed sulcation with white matter volume loss, inferior vermian hypoplasia |
| 473 | 6 | P | IUD | 21 w | Micro-anencephaly, ventriculomegaly, hydrops, pulmonary hypoplasia, inclusions in the lung, liver, kidney, pancreas and brain | Normal size, plasmacytic villitis, no inclusions | Unknown | HCMV DNA detected by PCR on amniocentesis |
| 660 | 6 | R | NND | 4 w | Splenomegaly, myocarditis, pneumonitis, hypoxic-ischaemic encephalopathy, inclusions in lung, pancreas | Not available | Unknown | Born at 35 weeks, intra-uterine growth retardation, out of hospital cardiac arrest, resuscitated, intensive therapy unit for 4 weeks |
| 68 | 7 | R | NND | 6 d | HCMV encephalitis and pneumonitis, inclusions in lung, kidney, ovary, adrenal, group B streptococcus pneumonia | Not available | Unknown | Normal pregnancy, normal growth |
| 124 | 7 | P, R | MC | 19 w | Mild hydrops, chronic stress, liver necrosis, myocarditis, encephalitis, inclusions in lung, liver, kidney and pancreas | Dichorionic diamniotic twin normal size; hydropic villi, avascular villi, focal plasmacytic villitiis, HCMV inclusions | Unknown | None stated |

[a]FFPE sample age (years) from collection to sequencing.

[b]P, placenta; R, kidney.

[c]IUD, intra-uterine death; TOP, termination of pregnancy; NND, neonatal death; MC, miscarriage.

[d]Gestation in weeks (w), or age in days (d) or weeks (w) for NND source.

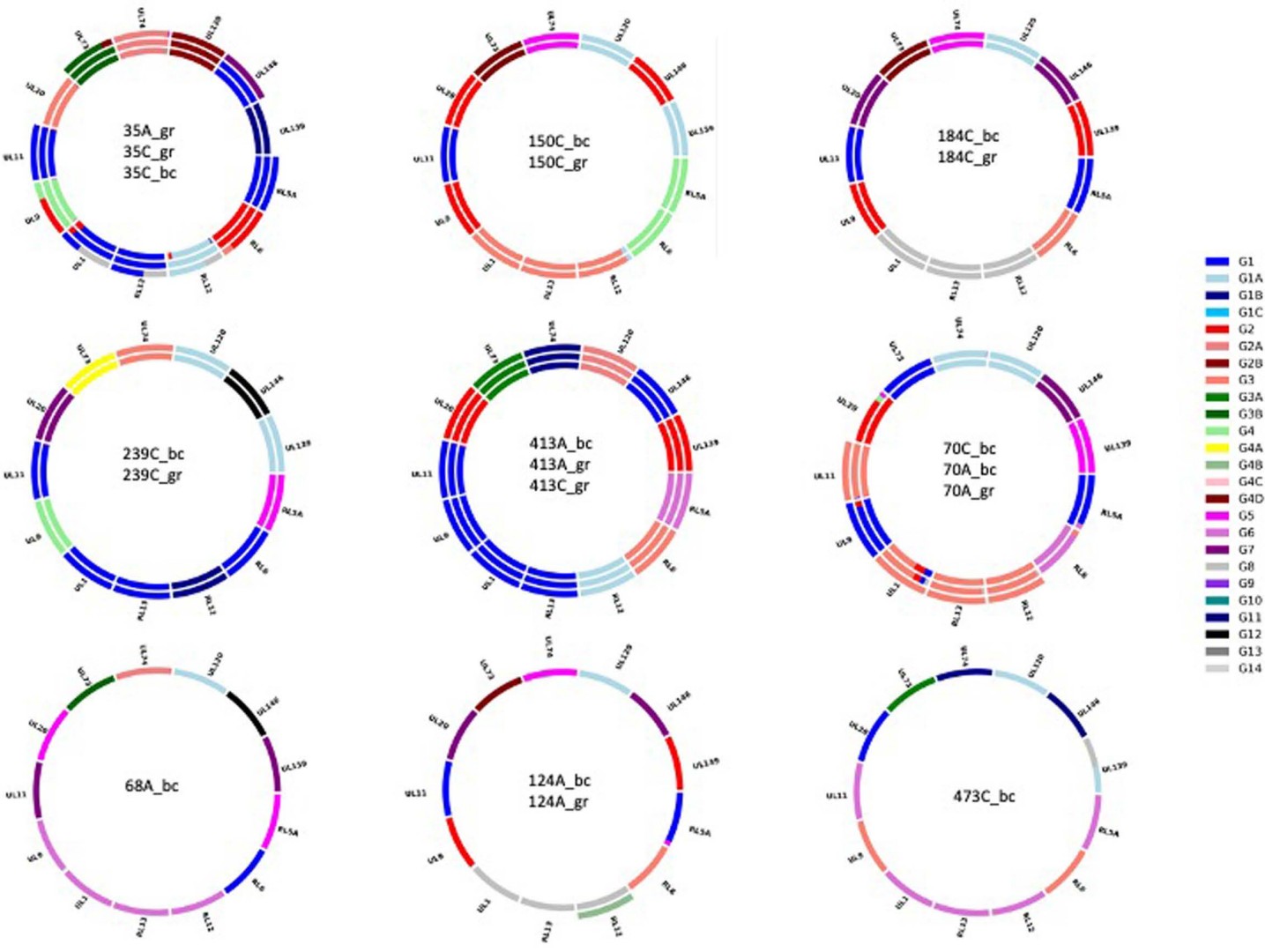

**Fig. 1. Doughnut plots reporting HCMV genotypes from dataset analysis.** Each ring represents an individual dataset, and is divided into sections representing the 13 hypervariable genes analysed. Datasets are listed from the outer ring inwards. The size of the coloured bars corresponds to the proportion of genotypes detected for each gene, as coded in the panel on the right using published genotype nomenclature (https://github.com/salvocamiolo/minion_Genotyper/blob/master/depositedSequences_codes.txt). Blank segments indicate that genotyping failed thresholds. Dataset names consist of the case number suffixed by P (placenta) or R (kidney) and then by _fp (FormaPure extraction kit) or _gr (GeneRead extraction kit).

Whole genome sequences were determined for five cases (Table 2) with relatively high HCMV load. The sequences from cases 413 and 239 exhibit unusual characteristics. The HCMV genome (236 kbp) has the structure $ab$-$U_L$-$b'a'c'$-$U_S$-$ca$, where $U_L$ and $U_S$ are long and short unique regions, respectively, flanked by inverted repeats $a$, $b$ and $c$ and their reverse complements $a'$, $b'$ and $c'$. For case 413, two versions (318 and 288 bp) of a subsequence of $c/c'$ were detected in approximately equal proportions. These versions may be present in a single genome population with one subsequence in $c$ and the other in $c'$, or they may be segregated into two populations with identical copies in $c$ and $c'$ in each. For case 239, the $a$ sequence at the left genome end differs from the $a'$ sequence internally, the latter consisting of two fused, dissimilar $a'$ sequences and the former being identical to one of these sequences except for 8 bp at one end. These

**Table 2. Coverage statistics and deposition data for read datasets and genome sequences.**

| Dataset name[a] | HCMV UL97[b] | Human FOX2P[b] | Original reads (no.)[c] | HCMV reads (%)[d] | Coverage (reads/nt) | SRA accession | GenBank accession[e] |
|---|---|---|---|---|---|---|---|
| 184P_fp | 11,911 | 672 | 16,364,538 | 85 | 8,724 | SRR31214615 | OR546128 |
| 184P_gr | 27,698 | 960 | 20,182,582 | 83 | 10,552 | SRR31214614 | |
| 70R_fp | 100,732 | 43,162 | 9,762,404 | 12 | 616 | SRR31214606 | |
| 70R_gr | 32,962 | 3,130 | 13,762,840 | 14 | 1,134 | SRR31214605 | |
| 150P_fp | 23,700 | 1,093 | 10,207,696 | 73 | 4,566 | SRR31214604 | OR546127 |
| 150P_gr | 17,726 | 939 | 17,007,560 | 27 | 2,844 | SRR31214603 | |
| 413P_fp | 50,974 | 21,992 | 12,841,786 | 9 | 739 | SRR31214602 | OR546130 |
| 413P_gr | 6,302 | 785 | 17,447,048 | 40 | 4,286 | SRR31214601 | |
| 413R_fp | 55,849 | 45,635 | 16,105,508 | 5 | 477 | SRR31214600 | |
| 413R_gr | 6,773 | 1,116 | 18,376,158 | 13 | 1,558 | SRR31214599 | |
| 35P_fp | 116,574 | 13,663 | 12,891,102 | 24 | 1,893 | SRR31214613 | OR546126 |
| 35P_gr | 12,693 | 525 | 21,232,492 | 40 | 5,297 | SRR31214612 | |
| 35R_gr | 745 | 562 | 13,331,384 | 2 | 142 | SRR31214611 | |
| 239P_fp | 11,204 | 2,722 | 18,438,992 | 3 | 365 | SRR31214610 | OR546129 |
| 239P_gr | 5,601 | 694 | 20,297,308 | 12 | 1,539 | SRR31214609 | |
| 473P_fp | 3,822 | 1,119 | 18,865,854 | 2 | 324 | SRR31214608 | |
| 68R_fp | 2,198 | 1,241 | 19,176,338 | 3 | 422 | SRR31214607 | |

[a]The case no. is suffixed by P (placenta) or R (kidney) and then by _fp (FormaPure extraction kit) or _gr (GeneRead extraction kit).

[b]IU/µL of HCMV UL97 or copies/µL human FOXP2 determined in the extracts by qPCR.

[c]Paired-end reads of 151 nt each.

[d]Read datasets were trimmed and aligned to the final sequence as described in the text. HCMV-related data are not included for case 124 because a final sequence was not obtained.

[e]Accessions relate to the HCMV genome represented in all read datasets from a case.

characteristics were present in both the placental and kidney samples from each case and were therefore unlikely to have been artefactual.

Variant calling identified 14 SNPs distributed among four cases (Table 3). All but one SNP was present in a single dataset at low frequency, and ten were C:G to T:A mutations, which occur in FFPE samples due to hydrolytic deamination of C residues to form U residues. Seven of the C:G to T:A mutations were detected in samples extracted using the FormaPure kit, which, unlike the GeneRead kit, does not incorporate uracil-DNA glycosylase to remove mismatched U residues. A single SNP was detected in both samples from case 239 at high frequency (≥36%).

## Discussion

This study met its objective by demonstrating that whole HCMV genomes may be sequenced from cCMV FFPE material. This was achieved with samples that had been archived for up to five years; it is possible that low HCMV load, rather than poor quality DNA, was the main contributor to low read coverage in older samples. Given the scarcity of fresh cCMV samples and the consequent small number and geographical restrictions of samples employed in published studies on the role of HCMV variation and strain composition in clinical outcome [2], this advance may result in FFPE repositories located worldwide proving key to future studies.

Ancillary data on the number of HCMV strains in the samples (by genotyping) and the occurrence of SNPs (by variant calling) were also obtained in this study, but, given the limitations mentioned above, conclusions relating to clinical outcome were not an objective. Future work would profit not only from the greater sample numbers that FFPE repositories

**Table 3. SNPs detected at levels over the threshold.**

| Dataset name[a] | Gene | Frequency (%) | Codon change | Amino acid change | C:G to T:A change |
|---|---|---|---|---|---|
| 184P_fp | UL85 | 5 | CCG to CCA | none | + |
| 150P_fp | UL52 | 6 | GCC to ACC | Ala to Thr | + |
| 150P_fp | UL57 | 7 | CGC to CAC | Arg to His | + |
| 413P_fp | RL13 | 5 | TGC to TGT | none | + |
| 413P_fp | UL16 | 6 | GCC to GCT | none | + |
| 413P_fp | UL54 | 6 | ACG to ATG | Thr to Met | + |
| 413P_gr | UL74 | 7 | ACA to ATA | Thr to Ile | + |
| 413P_gr | UL123 | 7 | AAG to AGG | Lys to Arg | − |
| 413P_gr | UL128 | 8 | GCG to TCG | Ala to Ser | − |
| 413P_gr | US24 | 8 | CCG to CCA | none | + |
| 413P_gr | US28 | 8 | GCC to GCT | none | + |
| 239P_fp | UL45 | 5 | GCT to GTT | Ala to Val | + |
| 239P_fp | UL147 | 36 | TAT to TGT[b] | Tyr to Cys | − |
| 239P_gr | UL147 | 38 | TAT to TGT[b] | Tyr to Cys | − |

[a]The case no. is suffixed by P (placenta) and then by _fp (FormaPure extraction kit) or _gr (GeneRead extraction kit).

[b]These SNPs refer to the same codon.

afford but also from investigating additional steps for preserving or repairing DNA integrity in FFPE material, with the objective of reducing the effects of formalin-induced artefacts on variant calling, and from side-by-side comparisons with fresh cCMV material.

## Supporting information

**S1 Table. Characteristics of extracts used to generate sequence datasets.**
(DOCX)

**S1 Fig. Plots characterising FFPE extracts prepared using the FormaPure or GeneRead kits and sequence data generated from these extracts.**
(DOCX)

## Acknowledgments

We thank Dr Phillip Cox, who was the consultant perinatal pathologist at Birmingham Women's Hospital, UK, and kindly provided pseudonymised cCMV FFPE samples.

## Author contributions

**Conceptualization:** Kathy K. Li, Andrew J. Davison, Richard J. Orton.

**Data curation:** Kathy K. Li, Andrew J. Davison, Richard J. Orton.

**Formal analysis:** Kathy K. Li, Andrew J. Davison, Richard J. Orton.

**Funding acquisition:** Kathy K. Li, Andrew J. Davison.

**Investigation:** Kathy K. Li, Richard J. Orton.

**Methodology:** Kathy K. Li, Nicolás M. Suárez, Salvatore Camiolo, Andrew J. Davison, Richard J. Orton.

**Project administration:** Kathy K. Li, Andrew J. Davison, Richard J. Orton.

**Resources:** Nicolás M. Suárez, Salvatore Camiolo.

**Software:** Salvatore Camiolo.

**Supervision:** Andrew J. Davison, Richard J. Orton.

**Writing – original draft:** Kathy K. Li, Andrew J. Davison.

**Writing – review & editing:** Kathy K. Li, Nicolás M. Suárez, Salvatore Camiolo, Andrew J. Davison, Richard J. Orton.

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
