## [Decision Letter · Decision Letter 0]

26 Feb 2025

PONE-D-25-02381DNA sequencing of whole human cytomegalovirus genomes from formalin-fixed, paraffin-embedded tissues from congenital cytomegalovirus disease casesPLOS ONE

Dear Dr. Li,

Thank you for submitting your manuscript to PLOS ONE. After careful consideration, we feel that it has merit but does not fully meet PLOS ONE’s publication criteria as it currently stands. Therefore, we invite you to submit a revised version of the manuscript that addresses the points raised during the review process.

Specifically, we want you to make the minor changes and clarifications suggested by the two experts who reviewed your manuscript.

We look forward to receiving your revised manuscript.

Kind regards,

Michael Nevels

Academic Editor

PLOS ONE

Journal Requirements:

For additional information about PLOS ONE ethical requirements for human subjects research, please refer to http://journals.plos.org/plosone/s/submission-guidelines#loc-human-subjects-research .

3. Please note that your Data Availability Statement is currently missing [the repository name and/or the DOI/accession number of each dataset OR a direct link to access each database]. If your manuscript is accepted for publication, you will be asked to provide these details on a very short timeline. We therefore suggest that you provide this information now, though we will not hold up the peer review process if you are unable.

Reviewers' comments:

Reviewer's Responses to Questions

**Comments to the Author**

1. Is the manuscript technically sound, and do the data support the conclusions?

Reviewer #1: Yes

Reviewer #2: Yes

2. Has the statistical analysis been performed appropriately and rigorously? 

Reviewer #1: Yes

Reviewer #2: N/A

3. Have the authors made all data underlying the findings in their manuscript fully available?

Reviewer #1: Yes

Reviewer #2: No

4. Is the manuscript presented in an intelligible fashion and written in standard English?

Reviewer #1: Yes

Reviewer #2: Yes

5. Review Comments to the Author

Reviewer #1: Li et al. present a study using FFPE tissue to perform whole HCMV genome sequencing. While the authors successfully genotyped nine and obtained whole genome sequences for five out of ten cCMV cases, demonstrating the potential of this approach for HCMV strain analysis from clinical FFPE samples, the manuscript requires further clarification. I have the following specific comments:

1. Most cCMV cases were diagnosed based on the post-mortem findings and histopathology, while it is not clear HCMV antigen was detected in the tissue in these patients.

2. The manuscript states (lines 126-127) that whole genome sequences were obtained for five cases with "relatively high HCMV load." However, the HCMV DNA load or UL97 read data for the remaining five cases, where whole genome sequencing was not successful, are not provided. This information is essential for evaluating the method's performance and would be valuable for future development and optimization.

Reviewer #2: Over all I believe the study is solid, however, limited in scope.

There is no statistical analysis, which in this case I belive is fine.

I think the DNA quality data is necessary to included in one form or another and study would benefit with a comparison of DNA quality and usability of the sample for WGS of HCMV genome.

The revies is written in an intelligible fashion of standard english.

The authors aims to assess the usability of archival FFPE tissue samples for whole genome sequence of human cytomegalovirus genomes. And they claim that this is indeed very feasible.

- Though this is not a very ambitious aim, I believe the presented data shows that this is possible.

- To improve the study, I think it would be more interesting to investigate what make a FFPE samples usable for the WGS of HCMV.

Sixteen FFPE samples from placental or kidney tissue of ten cases collected in the period of 2008-2018 are tested.

Two DNA extraction kits were used.

- I would like the authors to explain further why these two methods were used and comments on any differences were identified in DNA quality or downstream results? Is any of the kits better for special tissue?

The Authors find The DNA to be of sufficient quality for all but one sample.

- I would like to know how this quality is assessed and what the requirements is to be of sufficient quality. Please provide the DNA quality of all samples, at least as supplementary.

Three datasets were not meeting the threshold requirements of genotyping.

- Are there anything special with these data sets/ samples? How was the DNA quality?

For figure 1. The The abbriviations …”P_fp” and “R_gr” should be explained as they are in table 2.

The authors claim in the discussion that “it is possible that low HCMV load, rather than poor quality DNA, was the main contributor to low read coverage in older samples.”

- I would as mentioned before like to see the DNA quality data, and maybe do a direct comparison of HCMV load, DNA quality and quality of output data.

- It could also be interesting to look into storage time of the samples?

6. PLOS authors have the option to publish the peer review history of their article (what does this mean? ). If published, this will include your full peer review and any attached files.

**Do you want your identity to be public for this peer review?** For information about this choice, including consent withdrawal, please see our Privacy Policy .

Reviewer #1: No

Reviewer #2: No

---

## [Author Response · Author response to Decision Letter 1]

17 Apr 2025

Dear Editor,

Thank you for sending the reviewers’ comments. We reproduce them below, followed by our responses in red font in which line numbers refer to the revised manuscript. The relevant changes are also marked in red font in the revised manuscript. In addition to these improvements, we have made a few minor corrections that are not marked.

We have changed the corresponding author from Kathy K. Li to Richard J. Orton.

Reviewer #1: Li et al. present a study using FFPE tissue to perform whole HCMV genome sequencing. While the authors successfully genotyped nine and obtained whole genome sequences for five out of ten cCMV cases, demonstrating the potential of this approach for HCMV strain analysis from clinical FFPE samples, the manuscript requires further clarification. I have the following specific comments:

1. Most cCMV cases were diagnosed based on the post-mortem findings and histopathology, while it is not clear HCMV antigen was detected in the tissue in these patients. All the metadata provided to us is presented in Table 1. We have no further information on whether an antigen assay was applied to foetal tissues. However, all sequenced samples yielded HCMV data.

2. The manuscript states (lines 126-127) that whole genome sequences were obtained for five cases with "relatively high HCMV load." However, the HCMV DNA load or UL97 read data for the remaining five cases, where whole genome sequencing was not successful, are not provided. This information is essential for evaluating the method's performance and would be valuable for future development and optimization. Additional information is now included in Table S1 and Figure S1 [see also lines 74-77 and 107-111].

Reviewer #2: Over all I believe the study is solid, however, limited in scope.There is no statistical analysis, which in this case I belive is fine. I think the DNA quality data is necessary to included in one form or another and study would benefit with a comparison of DNA quality and usability of the sample for WGS of HCMV genome. Additional information is now included in Table S1 and Figure S1 [see also lines 74-77 and 107-111].

The revies is written in an intelligible fashion of standard english.The authors aims to assess the usability of archival FFPE tissue samples for whole genome sequence of human cytomegalovirus genomes. And they claim that this is indeed very feasible.- Though this is not a very ambitious aim, I believe the presented data shows that this is possible.- To improve the study, I think it would be more interesting to investigate what make a FFPE samples usable for the WGS of HCMV. This question is beyond the scope of our study, the stated objective of which was to sequence whole HCMV genomes from cCMV FFPE material. Indeed, it would be a more interesting question had the samples not proved usable.

Sixteen FFPE samples from placental or kidney tissue of ten cases collected in the period of 2008-2018 are tested.Two DNA extraction kits were used. I would like the authors to explain further why these two methods were used and comments on any differences were identified in DNA quality or downstream results? Is any of the kits better for special tissue? We have indicated that we chose two commercial kits that work via different methodologies [line 71]. Additional information is now included in Table S1 and Figure S1 [see also lines 74-77 and 107-111].

The Authors find The DNA to be of sufficient quality for all but one sample.- I would like to know how this quality is assessed and what the requirements is to be of sufficient quality. Please provide the DNA quality of all samples, at least as supplementary. Additional information is now included in Table S1 and Figure S1 [see also lines 74-77 and 107-111].

Three datasets were not meeting the threshold requirements of genotyping.- Are there anything special with these data sets/ samples? How was the DNA quality? Additional information is now included in lines 114-115 to indicate that this was probably due to a combination of low DNA load and low proportion of HCMV DNA load.

For figure 1. The The abbriviations …”P_fp” and “R_gr” should be explained as they are in table 2. This information has been added to the legend to Fig. 1 [lines 128-130].

The authors claim in the discussion that “it is possible that low HCMV load, rather than poor quality DNA, was the main contributor to low read coverage in older samples.” I would as mentioned before like to see the DNA quality data, and maybe do a direct comparison of HCMV load, DNA quality and quality of output data. Additional information is now included in Table S1 and Figure S1 [see also lines 74-77 and 107-111].

It could also be interesting to look into storage time of the samples? This information was included in Table 1. As is implicit in lines 169-170, we do not think that it is possible to draw valid conclusions on the viability of the two older samples, especially given their low HCMV loads.

---

## [Editor Report · Decision Letter 1]

22 Apr 2025

DNA sequencing of whole human cytomegalovirus genomes from formalin-fixed, paraffin-embedded tissues from congenital cytomegalovirus disease cases

PONE-D-25-02381R1

Dear Dr. Li,

We’re pleased to inform you that your manuscript has been judged scientifically suitable for publication and will be formally accepted for publication once it meets all outstanding technical requirements.

Kind regards,

Michael Nevels

Academic Editor

PLOS ONE
---

## [Editor Report · Acceptance letter]

PONE-D-25-02381R1

PLOS ONE

Dear Dr. Li,

I'm pleased to inform you that your manuscript has been deemed suitable for publication in PLOS ONE. Congratulations! Your manuscript is now being handed over to our production team.

Kind regards,

on behalf of

Dr. Michael Nevels

Academic Editor

PLOS ONE